# Alignment Attention by Matching
# Key and Query Distributions

**Shujian Zhang   Xinjie Fan   Huangjie Zheng   Korawat Tanwisuth   Mingyuan Zhou**
The University of Texas at Austin
{szhang19, xfan, huangjie.zheng, korawat.tanwisuth}@utexas.edu
mingyuan.zhou@mccombs.utexas.edu

## Abstract

The neural attention mechanism has been incorporated into deep neural networks to achieve state-of-the-art performance in various domains. Most such models use multi-head self-attention which is appealing for the ability to attend to information from different perspectives. This paper introduces alignment attention that explicitly encourages self-attention to match the distributions of the key and query within each head. The resulting alignment attention networks can be optimized as an unsupervised regularization in the existing attention framework. It is simple to convert any models with self-attention, including pre-trained ones, to the proposed alignment attention. On a variety of language understanding tasks, we show the effectiveness of our method in accuracy, uncertainty estimation, generalization across domains, and robustness to adversarial attacks. We further demonstrate the general applicability of our approach on graph attention and visual question answering, showing the great potential of incorporating our alignment method into various attention-related tasks.

## 1   Introduction

Attention-based mechanisms aggregate features with learnable weights to introduce useful inductive biases for sequence models [1, 2]. Since the introduction of the self-attention based Transformer [3], attention has become the foundation for many state-of-the-art models. Exploiting its computational efficiency and scalability, it has been used to train unprecedented large models on big datasets [4]. Large attention-based models have been demonstrating their ability to learn good representations in an unsupervised manner and benefit downstream analysis, with tremendous success in various natural language processing (NLP) [4–9], compute vision [10, 11], and multi-modal learning tasks [12, 13].

Attention networks, including multi-head attention, are being effectively utilized to capture the correlations between each pair of input tokens through individual or multiple attention functions. More specifically, in a self-attention layer with $H$ heads, assuming that the output of the previous layer consists of $n$ tokens, each of which is represented as a feature vector of dimension $d_{model} = d \cdot H$, then each token feature vector will be transformed by a $d_{model} \times d$ query projection matrix into a query feature vector, by a $d_{model} \times d$ key projection matrix into a key feature vector, and by a $d_{model} \times d$ value projection matrix into a value feature vector. The inner products of the $i$th query feature vector with all $n$ key feature vectors are then fed through a softmax function to define the relative weights of the $n$ keys to that query, which are used to aggregate the $n$ value vectors into the vector representation of the $i$th word token in a head.

Although such networks are simple to optimize and intuitive to understand, how the key and query projection matrices should differ from each other has not been well-studied and understood. It is thus

---

The code is available at `https://github.com/szhang42/alignment_attention`

35th Conference on Neural Information Processing Systems (NeurIPS 2021).

unclear whether they would result in well-controlled interactions between the keys and queries. In particular, ignoring the dependence between the $n$ token feature vectors, we can view the output of the previous layer as an empirical distribution supported on $n$ points in $\mathbb{R}^{d_{model}}$. In each head, this empirical distribution is transformed by the query and key projection matrices into a query empirical distribution and a key empirical distribution, respectively, each supported on $n$ points at the same feature space in $\mathbb{R}^d$. Since at each head two different projections matrices are used to project the input, the distributions of key and query will be different. Intuitively, if these two distributions are clearly misaligned with each other, then the query and key of a token, whose input feature resides in a region with lower probabilities, may have increased risk of being pushed further away from each other in the shared projection space.

This paper proposes alignment attention, which regularizes the query and key projection matrices at each self-attention layer, by matching the empirical distributions of the query and key feature vectors. We focus on within-head alignment between empirical distributions and present three different options for distribution matching. In our framework, alignment attention, built as an unsupervised approach to match the query and key distributions, is trained jointly to maximize a combination of data likelihood and distribution agreement. This efficient architecture design enables us to easily add alignment loss to convert existing self-attention networks, including pre-trained ones, into alignment attention. Meanwhile, it naturally shares parameters and computation with the self-attention networks, allowing an end-to-end training.

With a generic architecture, alignment attention can convert any existing soft attention models, including pre-trained ones, while maintaining the inherent advantages of conventional attention, such as efficiency and being simple to optimize. The proposed method boosts the performance while remaining efficient in memory and computation cost. Our experiments show that the proposed alignment attention method outperforms state-of-the-art self-attentions in a wide variety of settings, including natural language understanding tasks, graph attention network, and visual question answering, in terms of accuracy and uncertainty estimation. We further demonstrate that alignment attention achieves strong performance in domain generalization and adversarial robustness.

## 2  Alignment attention

We introduce a general recipe for alignment attention: (a) build the alignment to match the key and query distributions within each head, (b) develop efficient distribution matching methods, and (c) leverage existing attention structure and optimize the model in an end-to-end manner. The resulting architecture can be efficiently learned with existing self-attention networks.

### 2.1  Attention modules

Attention uses keys and queries to obtain soft attention weights $W$, which are then used to aggregate the values to obtain the output features. Consider $n$ key-value pairs with a key matrix $K \in \mathbb{R}^{n \times d_k}$, a value matrix $V \in \mathbb{R}^{n \times d_v}$, and $m$ queries $Q \in \mathbb{R}^{m \times d_k}$, where in general the dimensions of queries and keys are equal. The scaled product between key and query [3] is: $\Phi = f_{\text{dot}}(Q, K) = QK^T/\sqrt{d_k} \in \mathbb{R}^{m \times n}$. Alternative choices include dot-product [3, 4] and additive attention [14–16]. Attention weights $W$ is defined as the softmax output of $\Phi$: $W = \text{softmax}(\Phi)$, where $W_{i,j} = \frac{\exp(\Phi_{i,j})}{\sum_{j'=1}^{n} \exp(\Phi_{i,j'})}$ represents the importance of the $j$th key to the $i$th query learned by the neural networks.

The multi-head attention, first proposed in Transformer [3], projects the queries, keys, and values into $H$ subspaces with $H$ different learnable linear projections. These projections are performed in parallel and then concatenated into a single latent representation. At the $l$th self-attention layer, we can obtain attention weight $W^{l,h} = \text{softmax}(f(Q^{l,h}, K^{l,h}))$, where $Q^{l,h} = Q^l M_Q^{l,h}$, $K^{l,h} = K^l M_K^{l,h}$, and $V^{l,h} = V^l M_V^{l,h}$ for $h = 1, ..., H$, with $M$ denoting the parametric matrices to learn. The attention results from all heads are then concatenated into the layer output as $O^l = [W^{l,1}V^{l,1}, ..., W^{l,H}V^{l,H}]$.

### 2.2  Alignment attention

Self-attention allows the model to attend to the information from each representation subspace at each position [17]. To encourage different attention heads to indeed capture distinct features, most previous studies focus on the disagreement regularization to explicitly encourage the diversity

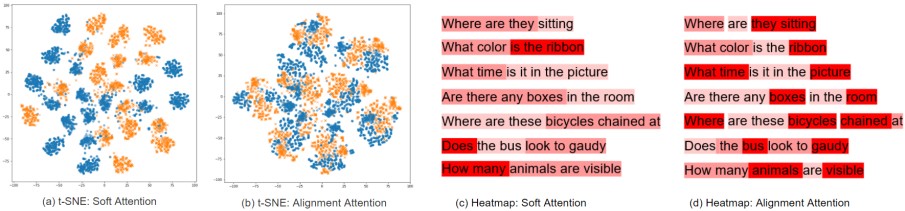

Figure 1: T-SNE visualizations of (a) soft attention and (b) alignment attention with bidirectional conditional transport. In (a) and (b), we visualize the both key distribution and the query distribution. The blue dots and orange diamonds represent the features from the key distribution and these from the query distribution, respectively. We show attention heatmaps of a VQA data in (c) and (d). It includes overall attention of seven examples' matrix embedding by summing up the attention weight vectors, with darker color denoting higher attention probability.

among attention heads [17]. By contrast, our focus is on exploiting how to make the key and query distributions better interact with each other in the latent space. We propose agreement matching to encourage the distributions of key and query over different tokens to be consistent within each head. Given a source input $\boldsymbol{x}$ and its output $\boldsymbol{y}$, a neural attention model is trained to maximize the conditional probability of $\boldsymbol{y}$ given $\boldsymbol{x}$ over a training corpus. We introduce distribution alignment, an auxiliary regularization term in order to encourage the alignment between the learned key and query distributions. Considering a supervised learning problem with training data $\mathcal{D} := \{\boldsymbol{x}_i, \boldsymbol{y}_i\}_{i=1}^N$, the likelihood parameterized by $\boldsymbol{\theta}$ is denoted by $p_{\boldsymbol{\theta}}(\boldsymbol{y}_i \,|\, \boldsymbol{x}_i)$. For notational convenience, below we drop the data index $i$. The whole model is differentiable to directly maximize the likelihood. Formally, the training objective with alignment attention is expressed as:

$$\mathcal{L}(\boldsymbol{x}, \boldsymbol{y}) = \underbrace{\log p_\theta(\boldsymbol{y} \mid \boldsymbol{x})}_{\text{likelihood}} + \lambda * \underbrace{\mathcal{L}_{\text{Align}}(Q, K)}_{\text{alignment}}, \tag{1}$$

where $\lambda$ is the alignment weight [18–21]. The auxiliary regularization term $\mathcal{L}_{\text{Align}}$ guides the distribution matching between key ($K$) and query ($Q$).

The proposed alignment provides a sample-and-head-dependent matching between the key and query. Assuming $Q$ and $K$ have the same batch size and number of heads; $Q$ is of dimension $[B, H, n, d_q]$, where $B$ represents the batch size, $H$ the number of heads, $n$ the number of queries, and $d_q$ the hidden dimension within a head; and $K$ is of dimension $[B, H, m, d_k]$. Focused on the self-attention networks, we assume $n = m = w$ and $d_q = d_k = d$ in our alignment attention. We use $Q, K$ to calculate the point-to-point difference from the query to key for each head at each sample, resulting in a tensor with dimension $[B, H, w, w]$. At each of the $B$ samples of the minibatch and each of the $H$ heads, the training objective is to minimize the expected difference between the empirical distributions of query and key, both of which are supported on a set of $w$ query/key features in $\mathbb{R}^d$ (see Figure 2). This flexible alignment method could be conveniently deployed into a single-head or multi-head attention mechanism.

With the alignment attention, the resulting key and query distributions should be close to each other (see t-SNE plots [22, 23]in Figure 1). We also visualize the attention weight from both soft and alignment attentions in Figure 1. It is clear that while many semantically and sentimentally important words and their combinations, such as "where, they, sitting," "color, ribbon," "what, time, picture," "boxes, room," "where, bicycles, chained," "bus, gaudy," and "animals, visible," are overlooked by vanilla soft attention, they are appropriately highlighted by the proposed alignment attention.

## 2.3 Alignment methods

To align the key and query distributions, we need a method that can quantify the difference between two distributions given their empirical samples. Under this requirement, we consider three different distribution matching methods, including the discriminator-based adversarial training [24], which is directly related to the Jensen–Shannon (JS) divergence [25], the Wasserstein distance in its primal form, which can be defined by solving an optimal transport problem [26, 27], and bi-directional conditional transport [28], which is developed by exploiting both the chain rule and Bayes' rule to quantify the difference between two probability distributions.

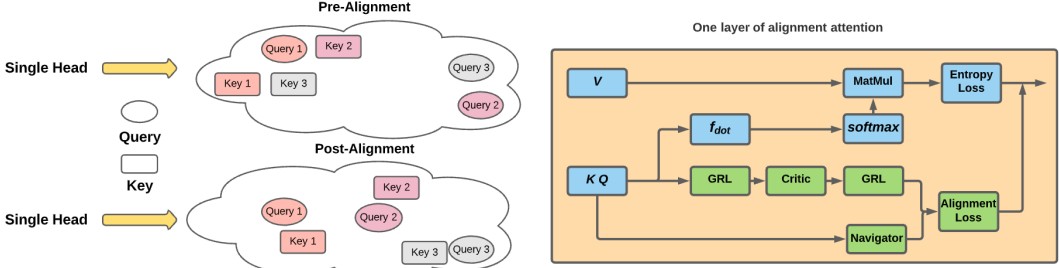

Figure 2: On the left, we visualize the alignment attention. The ellipse represents query and the rectangle represents the key. On the right, a demonstration of the difference and similarity between vanilla soft attention and our alignment attention. Alignment attention (in green) shares the same architecture as soft attention before obtaining key, query, and value. Then alignment attention adds the alignment structure to perform distribution matching, where GRL represents the gradient reversal layer.

### 2.3.1 Adversarial training-based alignment

Adversarial training, the key part to enable GANs [24], has been successfully exploited to minimize the distributional discrepancy [29, 30]. Under a minimax two-player framework, the discriminator $D$ is trained to distinguish the distributions of key and query, while both the query and key projection matrices, $M_Q$ and $M_K$, are treated as the generator $G$ and trained to confuse $D$. Here we consider a discriminator $D$ that is shared across the $H$ heads for alignment. Denote the empirical distributions of query and key as $p_Q$ and $p_K$, respectively. For a sample of $w$ tokens and at one of the $H$ heads, we have $p_Q = \sum_{i=1}^{w} \frac{1}{w} \delta_{q_i}$ and $p_K = \sum_{j=1}^{w} \frac{1}{w} \delta_{k_j}$, where $q_i$ and $k_i$ are projected from the $i$th token's input feature via the query and key projection matrices at that head, respectively. Thus the alignment loss is the sum over $H$ different head-dependent loss, each of which on a sample can be expressed as (we drop the head index for notational simplicity)

$$\min_{G} \max_{D} \mathcal{L}_{\text{Align-GAN}}(Q, K) := \mathbb{E}_{q \sim p_Q}[\log D(q)] + \mathbb{E}_{k \sim p_K}[\log(1 - D(k))]. \tag{2}$$

In summary, the discriminator is trained to maximize the alignment loss, and the generator, *i.e.*, the query and key projection matrices, is trained to minimize the alignment loss, so that the key and query distributions are learned adversarially to align with each other.

**Discriminator-based modules.** We leverage the highway architecture [31] to construct the discriminator. With a feature map $X \in \{Q, K\}$, the highway network first computes a transform gate $\tau$, as $\tau = \sigma(\Phi_\tau(X))$, where $\Phi_\tau$ is a fully-connected layer and $\sigma$ is the sigmoid activation function. Then, the output of the highway network is, $I_X = \tau \odot \text{ReLU}(\Phi_h(X)) + (1 - \tau) \odot X$, where $\Phi_h$ is another linear layer followed by ReLU activation and $\odot$ denotes the element-wise multiplication. We further apply a two-layer MLP to obtain the classifier probability, $D(X) = \sigma(\Phi_2(F_{\text{NL}}(\Phi_1(I_X))))$, where $\Phi_1$ and $\Phi_2$ are fully-connected layers connected by $F_{\text{NL}}$, a leaky ReLU activation function. To optimize the discriminator-based modules, instead of alternately updating the adversaries, like in GAN [24], we use the gradient-reversal layer [32] to jointly optimize all the components.

### 2.3.2 Optimal transport-based alignment

An alternative to the adversarial training-based alignment is to consider optimal transport (OT) [26], which has been widely used for distribution matching. In OT, the transport plan $\Pi(p_Q, p_K)$ is the set of all possible joint distributions of query and key, whose marginals over the query and key are $p_K$ and $p_Q$, respectively. We define the OT-based alignment cost as

$$\mathcal{L}_{\text{Align-OT}}(Q, K) = \min_{\pi \in \Pi(p_Q, p_K)} \mathbb{E}_{q, k \sim \pi}[c(q, k)], \tag{3}$$

where $c(\cdot, \cdot)$ is the transport cost between two points. For discrete $p_Q$ and $p_K$, fining the optimal transport plan often involves solving a computationally expensive linear programming problem. It is also known in the literature that OT is sensitive to the outliers in $p_Q$ and $p_K$ due to the two marginal constraints [33]. To reduce the computation burden, we consider an entropy-regularized OT, which allows the OT problem to be solved iteratively using the Sinkhorn iterations [34].

**Optimal transport-based modules.** We use the Sinkhorn algorithm [34, 35] to estimate the transport plan, defined as $\pi_\epsilon = \underset{\pi}{\mathrm{argmin}} \sum_{i=1}^{w} \sum_{j=1}^{w} (c(q_i, k_j) \cdot \pi(q_i, k_j) - \epsilon \pi(q_i, k_j) \log \pi(q_i, k_j))$, where $\epsilon = 0.01$. The OT-alignment cost now becomes $\mathcal{L}_{\mathrm{Align-OT}} = \sum_{i=1}^{w} \sum_{j=1}^{w} (c(q_i, k_j) \cdot \pi_\epsilon(q_i, k_j))$.

### 2.3.3 Bidirectional conditional transport-based alignment

Instead of requiring $\pi \in \Pi(p_Q, p_K)$ as in OT, we follow Zheng and Zhou [28] to consider probabilistic bidirectional conditional transport (CT), which exploits both the chain rule and Bayes' theorem to constrain the joint $\pi$ with both $p_Q = \sum_{i=1}^{w} \frac{1}{w} \delta_{q_i}$ and $p_K = \sum_{j=1}^{w} \frac{1}{w} \delta_{k_j}$ in two different ways. First, we consider a query-to-key CT that define $\pi$ as $\pi(q_i, k_j) = p_Q(q_i) \pi_K(k_j \mid q_i)$, where $\pi_K(k_j \mid q_i)$ is a conditional distribution defined as $\pi_K(k_j \mid q_i) = \frac{p_K(k_j) \exp(\tau_\phi(k_j)^T \tau_\phi(q_i))}{\sum_{j'=1}^{w} p_K(k_{j'}) \exp(\tau_\phi(k_{j'})^T \tau_\phi(q_i))}$ and $\tau_\phi(\cdot)$ is a neural network based transformation parameterized by $\phi$. Second, we consider a key-to-query CT that defines $\pi(q_i, k_j) = p_K(k_j) \pi_Q(q_i \mid k_j)$, where $\pi_Q(q_i \mid k_j) = \frac{p_Q(q_i) \exp(\tau_\phi(k_j)^T \tau_\phi(q_i))}{\sum_{i'=1}^{w} p_Q(q_{i'}) \exp(\tau_\phi(k_j)^T \tau_\phi(q_{i'}))}$. Third, we define a point-to-point cost as $c_\eta(q, k) = 1 - \frac{\tau_\eta(k)^T \tau_\eta(q)}{||\tau_\eta(k)||_2 ||\tau_\eta(q)||_2}$, where $\tau_\eta(\cdot)$ is a neural network based "critic" function whose parameter $\eta$ will be adversarially learned. Combing them together leads to the bidirectional CT-based alignment loss as

$$\mathcal{L}_{\mathrm{Align-CT}}(Q, K) = \tfrac{1}{2} \mathbb{E}_{q \sim p_Q, \, k \sim \pi_K(\cdot \mid q)} [c_\eta(q, k)] + \tfrac{1}{2} \mathbb{E}_{k \sim p_K, \, q \sim \pi_Q(\cdot \mid k)} [c_\eta(q, k)]. \qquad (4)$$

Compared to OT, bidirectional CT is able to efficiently model the alignment with less computation. The structure of alignment attention with transport-based methods is presented in Fig. 2.

**CT-based modules.** The critic $\tau_\eta(\cdot)$ is structured similarly as the discriminator in Section 2.3.1. It projects the input data onto a vector space instead of outputting logits for binary classification. For $\tau_\phi(\cdot)$, we use a two-layer MLP network. We optimize the query and key projection matrices and $\phi$ to minimize $\mathcal{L}_{\mathrm{Align-CT}}(Q, K)$ in (4), and optimize $\eta$ to maximize it. The gradient-reversal layer [32] is also used to optimize the critic adversarially.

## 3 Related work

**Alignment Learning.** Liang et al. [36] first assign agreement terms for jointly training word alignment in phrase-based statistic machine translation. The general bidirectional sequence alignment models with model inevitability regularization are then proposed by Levinboim et al. [37]. Recently, the alignment is studied in the multi-head attention model based on the Transformer architecture, where alignment extraction is improved by augmenting an additional alignment head to the multi-head source-to-target attention component [38]. Our proposed alignment attention adopts the alignment idea and matches the distributions of query and key. This general and efficient framework gives us the flexibility to better model attention weights. The domain generalization ability and adversarial robustness of alignment attention are also studied.

**Distribution Matching.** Distribution matching is a fundamental problem in statistics and machine learning [39]. The widely used distances include the Kullback–Leibler (KL) divergence [40] and Jensen–Shannon (JS) divergence [25]. GANs [24] are proposed with the adversarial-based objectives. Due to the inherent advantages of allowing the two distributions to have non-overlapping supports [41, 42], the Wasserstein distance from the optimal transport problem is also used for defining the transport cost [26, 27]. In our alignment attention, we consider discriminator-based and transport-based methods as alignment methods. Based on the alignment methods, we apply alignment attention to the widely used attention models and leverage the existing efficient attention architecture to build entire alignment attention network.

## 4 Experiments

Our method can be incorporated into any self-attention based models. To exam its effectiveness and general applicability, we apply alignment attention to a diverse set of tasks, including language understanding, graph attention, and visual question answering. Furthermore, the model's generalization

across domains and robustness towards adversarial attacks are studied on language tasks. We study a variety of state-of-the-art models for these tasks including ALBERT [5], BERT [4], and RoBERTa [6]. Below we present the main experimental settings and results, with more training and hyperparameter details provided in Appendix B.

## 4.1 Alignment attention in natural language understanding

Since the self-attention based Transformer was proposed, it has been widely used in pretrained models for NLP and related areas, achieving state-of-the-art results on various downstream tasks. However, training a state-of-the-art pretrained model now requires substantial computational resources which demands considerable energy, along with the associated financial and environmental costs. Vaswani et al. [3] report that a Transformer-base model was trained on 8 Nvidia P100 GPUs for 12 hours and Strubell et al. [43] report a BERT-base model was trained on 64 V100 GPUs for 79 hours. Thus for alignment attention, we utilize it to finetune these pretrained models on large corpora, which is not only computationally and financially friendly but also accessible to more researchers.

### 4.1.1 In-domain language understanding evaluation

We first evaluate the alignment attention for in-domain language tasks where the training and testing data are from the same domain. We conduct experiments on eight benchmark datasets from General Language Understanding Evaluation (GLUE) [44] and two Stanford Question Answering Datasets (SQuAD) [45, 46]. Our experiments are based on the state-of-the-art pretrained model, ALBERT [5], a memory-efficient version of BERT [4] with parameter sharing and embedding factorization. Based on Huggingface PyTorch Transformer [47], our implementation uses the base version of ALBERT following the same setting from Lan et al. [5].

Table 1: Performance of alignment attention on GLUE and SQuAD benchmarks.

|  | MRPC | CoLA | RTE | MNLI | QNLI | QQP | SST | STS | SQuAD 1.1 | SQuAD 2.0 |
|---|---|---|---|---|---|---|---|---|---|---|
| ALBERT-base | 86.5 | 54.5 | 75.8 | 85.1 | 90.9 | 90.8 | 92.4 | 90.3 | 80.86/88.70 | 78.80/82.07 |
| ALBERT-base+**AA-GAN** | 87.5 | 55.7 | **77.3** | 85.8 | **91.3** | 91.4 | 92.6 | 91.1 | 81.19/88.92 | 79.25/82.57 |
| ALBERT-base+**AA-OT** | 87.9 | 54.6 | 77.0 | 85.7 | 91.2 | 91.3 | 92.8 | 91.2 | 81.13/88.89 | 79.18/82.48 |
| ALBERT-base+**AA-CT** | **88.6** | **55.9** | 77.2 | **85.9** | **91.3** | **91.5** | **93.1** | **91.5** | **81.32/89.02** | **79.33/82.71** |

**Results.** In Table 1, we present the soft attention and the alignment attention (AA) with GAN, optimal transport, and CT, resuming from the same checkpoints. The mean accuracies are reported with 5 independent runs (see full results with the error bars in Table 8 in the Appendix). Alignment attention outperforms the soft attention, which indicates that matching the distributions of key and query gives better performance than soft attention and the results are not sensitive to the choice of alignment methods. Due to the expensive computation for distance measure at each step of OT, we will focus on GAN and CT as alignment methods in the following experiments. Overall, alignment attention improves the soft attention in both GLUE and SQuAD datasets even by only using alignment attention at the finetuning stage.

### 4.1.2 Generalization across domains

In real applications, it is very common to deploy a neural network model into a new domain with data unseen during training. The model's generalization has been extensively studied in the machine learning community. Significant past work has studied cross-domain robustness using sentiment analysis [48–50]. The recent work from Desai and Durrett [51] has explicitly elected tasks where out-of-domain performance is substantially lower and challenging domain shifts are exhibited. Following the setting in Desai and Durrett [51], we test the generalization ability of our alignment attention. For our in-domain and out-of-domain datasets, we split the development set in half to obtain a held-out, non-blind test set. We conduct experiments on three tasks: (1) *Natural Language Inference.* The Stanford Natural Language Inference (SNLI) corpus is a large-scale entailment dataset [52] as the in-domain data. Multi-Genre Natural Language Inference (MNLI) [53] can be used as unseen out-of-domain test dataset. (2) *Paraphrase Detection.* Quora Question Pairs (QQP) is used as in-domain data which includes sentence pairs from Quora that are semantically equivalent [54]. TwitterPPDB (TPPDB) [55] is considered as out-of-domain data. (3) *Commonsense Reasoning.* Situations With Adversarial Generations (SWAG) is a grounded commonsense reasoning task [56]. The out-of-domain data is HellaSWAG (HSWAG), which is a more challenging benchmark [56]. We report accuracy and expected calibration error (ECE) for both in-domain(ID) and out-of-domain(OD).

ECE is calculated as a weighted average of the difference between each bin's accuracy and confidence: $\text{ECE} := \sum_i \frac{G_i}{N} |\text{acc}(G_i) - \text{conf}(G_i)|$, where $G_i$, $\text{acc}(G_i)$, and $\text{conf}(G_i)$ are the count, accuracy, and confidence of samples in the $i$'th group, respectively. The number of groups is 10 as in [51].

Table 2: Results of domain generalization. We report the accuracy and ECE of various models on both in-domain data and out-of-domain data for three tasks: natural language inference, paraphrase detection, and commonsense reasoning.

| | ACCURACY ↑ | | ECE ↓ | |
| --- | --- | --- | --- | --- |
| | ID | OD | ID | OD |
| NATURAL LANGUAGE INFERENCE | SNLI | MNLI | SNLI | MNLI |
| DA [57] | 84.63 | 57.12 | **1.02** | 8.79 |
| ESIM [58] | 88.32 | 60.91 | 1.33 | 12.78 |
| BERT-BASE [51] | 90.04 | 73.52 | 2.54 | 7.03 |
| BERT-BASE+**AA-GAN** | 90.59 | 74.15 | 2.02 | 5.82 |
| BERT-BASE+**AA-CT** | **90.65** | **74.23** | 1.89 | **5.65** |
| ROBERTA-BASE | 91.23 | 78.79 | **1.93** | 3.62 |
| ROBERTA-BASE+**AA-GAN** | 91.52 | 79.55 | 2.70 | 3.31 |
| ROBERTA-BASE+**AA-CT** | **91.68** | **79.60** | 2.52 | **2.79** |
| PARAPHRASE DETECTION | QQP | TWITTER | QQP | TWITTER |
| DA [57] | 85.85 | 83.36 | 3.37 | 9.79 |
| ESIM [58] | 87.75 | 84.00 | 3.65 | 8.38 |
| BERT-BASE [51] | 90.27 | 87.63 | 2.71 | 8.51 |
| BERT-BASE+**AA-GAN** | **90.80** | **88.34** | **1.45** | **7.48** |
| BERT-BASE+**AA-CT** | 90.62 | 88.25 | 1.74 | 7.52 |
| ROBERTA-BASE [51] | 91.11 | 86.72 | 2.33 | 9.55 |
| ROBERTA-BASE+**AA-GAN** | **91.66** | 87.28 | **1.78** | 9.45 |
| ROBERTA-BASE+**AA-CT** | 91.53 | **87.33** | 1.89 | **9.40** |
| COMMONSENSE REASONING | SWAG | HSWAG | SWAG | HSWAG |
| DA [57] | 46.80 | 32.48 | 5.98 | 40.37 |
| ESIM [58] | 52.09 | 32.08 | 7.01 | 19.57 |
| BERT-BASE [51] | 79.40 | 34.48 | 2.49 | 12.62 |
| BERT-BASE+**AA-GAN** | 79.56 | 35.90 | 1.95 | 12.11 |
| BERT-BASE+**AA-CT** | **79.60** | **36.25** | **1.86** | **11.78** |
| ROBERTA-BASE [51] | 82.45 | 41.68 | 1.76 | 11.93 |
| ROBERTA-BASE+**AA-GAN** | 83.03 | 42.51 | 1.61 | 9.97 |
| ROBERTA-BASE+**AA-CT** | **83.14** | **42.88** | **1.43** | **9.77** |

**Results.** In Table 2, we include open-source implementations of Decomposable Attention (DA) [57] and Enhanced Sequential Inference Model (ESIM) [58] as baselines. For pretrained models, we use BERT-base-uncased [4] and RoBERTa-base [6] from HuggingFace Transformers [47]. We incorporate the alignment attention in both pretrained models. Alignment attention consistently outperforms the corresponding soft attention not only for in-domain, confirming our results in Section 4.1.1, but also for out-of-domain. The relevantly larger gains on the out-of-domain setting indicate that alignment attention has the better generalization ability across domains. The improved results of ECE demonstrate the better-calibrated model for uncertainty estimation with alignment attention.

### 4.1.3 Robustness towards adversarial attacks

Machine learning models recently have been found vulnerable to adversarial examples that are legitimate input altered by small and often imperceptible perturbations [59]. Therefore, it becomes increasingly important to exam the model's robustness against adversarial attacks. Our alignment attention imposes a regularization to ensure well-aligned key and query distributions so that it is expected to become more robust to the generated perturbation that would fool the model. We follow the same settings from Section 4.1.1 to test the robustness of finetuned ALBERT-base models with soft attention or alignment attention. We utilize the TextAttack [60] framework and apply three state-of-the-art untargeted black-box adversarial attacks (1) Textfooler [61]: counter-fitted word embedding swap; (2) Textbugger [62]: character-level insertion, deletion, swap, and substitution; (3) BAE [63]: generating BERT masked token prediction. 1000 adversarial attacks are conducted for each model with a maximum sentence length of 512. We report the percentages of failed adversarial attacks in Table 3. Higher percentages indicate more robust models.

**Results.** The alignment attention shows consistent improvements over the soft attention across all three attacks and achieves significant gains on the average failure rates. The alignment attention

Table 3: Results of pretrained large-scale models' robustness against adversarial attacks. We report the percentages of failed attacks under three adversarial attacks respectively.

| ATTACK | ATTENTION | MRPC | COLA | RTE | QQP | SST-2 | AVG. |
|---|---|---|---|---|---|---|---|
| | BASE | 6.5 | 2.6 | 16.2 | 25.4 | 7.0 | 11.5 |
| TEXTFOOLER | AA-GAN | 8.4 | **7.1** | 14.2 | 31.2 | **14.5** | 15.1 |
| | AA-CT | **8.7** | 6.9 | **15.3** | **32.1** | 13.9 | **15.4** |
| | BASE | 10.6 | 16.8 | 19.9 | 30.1 | 40.1 | 23.5 |
| TEXTBUGGER | AA-GAN | 12.6 | **22.4** | 20.7 | 36.0 | 56.1 | 29.6 |
| | AA-CT | **13.1** | 20.9 | **21.0** | **36.5** | **57.6** | **29.8** |
| | BASE | 44.8 | 4.9 | 35.6 | 48.8 | 13.9 | 29.6 |
| BAE | AA-GAN | 44.2 | **7.3** | 35.8 | 46.5 | **17.8** | 30.3 |
| | AA-CT | **45.3** | 6.7 | **36.3** | **49.4** | 17.5 | **31.0** |

demonstrates its robustness and gets along with our intuition that the alignment attention can learn better and more robust key and query distribution due to the use of distributional matching.

## 4.2 Alignment attention in graph neural networks

To test the general applicability of our alignment attention, we also experiment our method with graph attention networks (GAT) [64], where the graph structure is injected into the attention masks in which nodes are able to attend over their neighborhoods' features in the graph. Leveraging masked self-attentional layers, GAT processes the node features for the node classification.

**Experimental Setup.** Following the setting in GAT [64], we conduct experiments on three standard citation network benchmark datasets— Cora, Citeseer and Pubmed [65]— in a transductive setting, indicating all nodes from training and test are on the same graph [66]. The details of three datasets and experimental settings are deferred to Appendix B.

Table 4: Classification accuracy for graphs.

| Attention | Cora | Citeseer | PubMed |
|---|---|---|---|
| GAT | 83.00 | 72.50 | 77.26 |
| **AA-GAN** | 83.78$\pm$0.2 | 73.32$\pm$0.1 | 78.77$\pm$0.2 |
| **AA-CT** | **83.80**$\pm$0.3 | **73.49**$\pm$0.2 | **78.79**$\pm$0.2 |

**Results.** In Table 4, we report the mean classification accuracies on test nodes over 5 random runs, and the standard deviations of alignment attention. We experiment with both AA-GAN and AA-CT. Table 4 shows that alignment attention consistently improves upon the corresponding baseline models across all three datasets, which further confirms the efficient structure of this alignment attention. The CT-based alignment attention performs better than GAN-based alignment attention.

## 4.3 Attention in visual question answering

Visual question answering (VQA) [67] is a multi-modal learning task where the model predicts the answer given a question conditioning on the image. Self-attention architectures in MCAN [68] has been recently proposed to learn the fine-grained semantic meaning of both the image and question. We adapt the proposed alignment attention to MCAN and compare with soft attention. We conduct experiments on the VQA-v2 dataset [67] and follow the hyperparameters and other settings from Yu et al. [68]. In addition, to investigate the model's robustness to noise, we construct the noisy dataset by incorporating the Gaussian noise (mean 0, variance 1) to image features [69, 70]. Four-layer encoder-decoder based MCAN is used as the baseline model with the soft self-attention. For each experiment, we report the accuracy on both the original data and the noisy data. As in Fan et al. [70] and Zhang et al. [71], we also report the Patch Accuracy vs Patch Uncertainty (PAvPU) [70, 72] as a measure of the uncertainty estimation where the $p$-value threshold is set to be $0.05$ and the number of attention weight samples is 20. Please refer more detailed experimental settings in Appendix B.

Table 5: Accuracies and PAvPUs of different attentions on both the original VQA-v2 dataset and the noise ones.

| | ACCURACY $\uparrow$ | | PAvPU $\uparrow$ | |
|---|---|---|---|---|
| | ORIGINAL | NOISY | ORIGINAL | NOISY |
| BASE | 66.74 | 63.58 | 71.96 | 68.29 |
| **AA-GAN** | 66.92 | 64.28 | 72.17 | 69.80 |
| **AA-CT** | **67.01**$\pm$0.02 | **64.57**$\pm$0.03 | **72.21**$\pm$0.03 | **69.98**$\pm$0.04 |

**Results.** The results are summarized in Table 5. We report the accuracy and uncertainty of different attentions on both original and noisy data. In terms of accuracy, alignment attention shows consistent improvements over the soft attention on both original and noisy data. These results verify our conjecture that the alignment attention is more robust to the noise which aligns with our results on adversarial robustness in Section 4.1.3. For uncertainty, we observe that on both original and noisy data, alignment attention has better uncertainty estimations, meaning that alignment attention in general is more certain on its correct predictions and more uncertain on its mistakes.

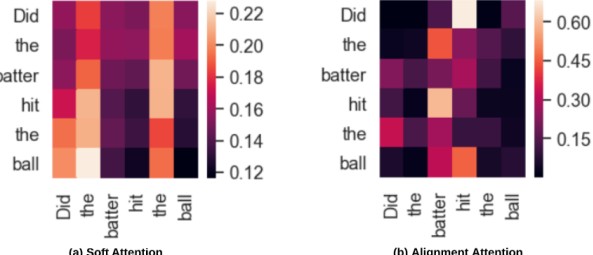

Figure 3: For one question from VQA, we visualize the attention weights of AA (a) and soft attention (b). Rows represent queries, and columns represent keys. For example, on the left plot, when the row is 'Did' and the column is 'hit', the color represents the average attention weight from the query 'Did' to the key 'hit'.

**Results Analysis.** *Visualizations.* We plot the attention weights of both alignment attention and soft attention on one question for VQA. In Figure 3, attention weight represents the average importance of each query and key pair. For example, in (b), when the row is 'Did' and the column is 'hit', the color represents the average attention weight from the query 'Did' to the key 'hit'. We observe that the alignment attention gives relatively sharper attention weights compared to the soft attention and therefore gives good prediction accuracy and uncertainty estimation.

**Parameter Size and Running time.** In Table 6, we provide the parameter sizes and step time for different attention types combined with MCAN where the attention module constructs the main model. It shows that alignment attention (AA) keeps the parameter size at the very similar level as soft attention while moderately increasing the step time compared to soft attention.

Table 6: Efficiency on VQA task.

| ATTENTION | PARAMS ↓ | S/STEP ↓ |
|---|---|---|
| BASE | 43.3M | 0.25 |
| **AA-GAN** | 43.4M | 0.31 |
| **AA-CT** | 43.4M | 0.36 |

**Ablation Study.** We conduct ablation study with AA+CT to exam the role of the alignment-weight hyperparameter $\lambda$ in Equation 1 by turning it from $0.01$ to $1$. We find that the experimental results are not sensitive to the choice of the value of the $\lambda$. Any number from $0.01$ to $1$ would give similar results. In all experiments considered in the paper, which cover various noise levels and model sizes, we have simply fixed it as $0.01$. Please see detailed results in Table 9 in Appendix.

## 5   Conclusion

Our proposed alignment attention aims to match the key and query distributions. We leverage different alignment methods with a generic and efficient architecture design which requires surprisingly few modifications to standard soft attention and enables us to easily convert existing soft attention models, including pretrained ones, to alignment attention. Our experiments on a variety of language under-standing tasks show that alignment attention achieves strong performance in accuracy, uncertainty estimation, domain generalization, and adversarial robustness even by only adding the alignment loss during the finetuning stage. In the real-life scenarios, the attention models have been deployed in many machine learning systems, such as self-driving [73] and healthcare [74]. However, the data from the real practice is biased and long-tailed. Therefore, we see opportunities of our method that can mitigate the risks with uncertainty estimation. Further, on graph node classification and visual question answering, alignment attention demonstrates its general applicability and effectiveness of

each component of the proposed structure, showing the great potential to be added as a plug-and-play component to many existing attention models.

## Acknowledgements

The authors acknowledge the support of Grant IIS-1812699 from the U.S. National Science Foundation, the APX 2019 project sponsored by the Office of the Vice President for Research at The University of Texas at Austin, the support of a gift fund from ByteDance Inc., and the Texas Advanced Computing Center (TACC) for providing HPC resources that have contributed to the research results reported within this paper.

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
