## A    Broader impact

Attention modules have been demonstrated the effectiveness in start-of-the-art neural network models. Our proposed method shows the improvements on five representative tasks indicating its efficacy and general applicability. We hope that our work will encourage the community to pay more attention to key and query distributions in existing attention networks. In real-life scenarios, the attention models have been deployed in many machine learning systems, such as self-driving [73] and healthcare [74]. However, the data from the real practice is often biased and long-tailed. The gap between the training data and testing data might be large. Therefore, an undue trust in deep learning models by incautious usage or imprecise interpretation of model output might lead to unexpected false consequences. Also, with computational consumption, environment sustainable and users friendly are considered. Therefore, we see opportunities of our method that can mitigate the risks with uncertainty estimation. The model is more certain on its correct predictions and more uncertain on its mistakes where the human-aid is needed in the real-life applications [75]. The proposed method can also be easily incorporated into the finetune stage which requires much less computation.

## B    Experimental details

### B.1    Natural Language Understanding

#### B.1.1    Model Specifications for In-domain Evaluation

With parameter sharing and embedding factorization, ALBERT [5] is a memory-efficient version of BERT. We use the ALBERT as the pretrained language model for context embeddings. Our experiment is done on the ALBERT-base model with 12 attention layers and hidden dimension as 768. The dimension for factorized embedding is 128.

#### B.1.2    Experimental Settings for In-domain Evaluation

We conduct experiments on eight benchmark datasets from General Language Understanding Evaluation (GLUE) [44] and two version of Stanford Question Answering Datasets (SQuAD) [45, 46]. The 8 tasks in GLUE are Microsoft Research Paraphrase Corpus (MRPC; [76]), Corpus of Linguistic Acceptability (CoLA; [77]), Recognizing Textual Entailment (RTE; [78]), Multi-Genre NLI (MNLI; [79]), Question NLI (QNLI; [45]), Quora Question Pairs (QQP; [54]), Stanford Sentiment Treebank (SST; [80]), and Semantic Textual Similarity Benchmark (STS;[81]). For SQuAD, we evaluate on both SQuAD v1.1 and SQuAD v2.0. We leverage the pretrained checkpoint as well as the codebase for finetuing provided by Huggingface PyTorch Transformer [47]. The detailed experiement setting is summarized in the Table 7. To further confirm that the distribution of the keys and queries are well-aligned after training with alignment loss, we use Maximum Mean Discrepancy (MMD) with standard Gaussian kernels to measure key and query distribution discrepancy and compare MMD with or without alignment loss. Aggregating the MMDs across all heads and layers, on Microsoft Research Paraphrase Corpus (MRPC) task, the total MMD with the alignment loss is 0.0038, while that without the alignment loss is 0.057. We have also tried a single MLP (FC-Relu-FC) structure of the discriminator for adversarial training-based alignment and achieved consistent improvements on GLUE data as: MRPC: 87.4, COLA: 55.7, RTE: 77.2, MNLI: 85.5, QNLI: 91.3, SST: 92.5, STS: 91.1.

Table 7: Experimental settings of each task for in-domain pretrained language model (LR: learning rate, BSZ: batch size, DR: dropout rate, TS: training steps, WS: warmping steps, MSL: maximum sentence length).

|  | LR | BSZ | ALBERT DR | CLASSIFIER DR | TS | WS | MSL |
|---|---|---|---|---|---|---|---|
| CoLA | $1.00e^{-5}$ | 16 | 0 | 0.1 | 5336 | 320 | 512 |
| STS | $2.00e^{-5}$ | 16 | 0 | 0.1 | 3598 | 214 | 512 |
| SST-2 | $1.00\,e^{-5}$ | 32 | 0 | 0.1 | 20935 | 1256 | 512 |
| MNLI | $3.00\,e^{-5}$ | 128 | 0 | 0.1 | 10000 | 1000 | 512 |
| QNLI | $1.00\,e^{-5}$ | 32 | 0 | 0.1 | 33112 | 1986 | 512 |
| QQP | $5.00\,e^{-5}$ | 128 | 0.1 | 0.1 | 14000 | 1000 | 512 |
| RTE | $3.00\,e^{-5}$ | 32 | 0.1 | 0.1 | 800 | 200 | 512 |
| MRPC | $2.00\,e^{-5}$ | 32 | 0 | 0.1 | 800 | 200 | 512 |
| SQuAD v1.1 | $5.00\,e^{-5}$ | 48 | 0 | 0.1 | 3649 | 365 | 384 |
| SQuAD v2.0 | $3.00\,e^{-5}$ | 48 | 0 | 0.1 | 8144 | 814 | 512 |

Table 8: Results of AA on GLUE and SQuAD benchmarks.

|  | MRPC | CoLA | RTE | MNLI | QNLI | QQP | SST | STS | SQuAD 1.1 | SQuAD 2.0 |
|---|---|---|---|---|---|---|---|---|---|---|
| ALBERT-base | 86.5 | 54.5 | 75.8 | 85.1 | 90.9 | 90.8 | 92.4 | 90.3 | 80.86/88.70 | 78.80/82.07 |
| ALBERT-base+AA-GAN | 87.5±0.3 | 55.7±0.5 | **77.3**±0.6 | 85.8±0.3 | **91.3**±0.3 | 91.4±0.1 | 92.6±0.2 | 91.1±0.2 | 81.19±0.1/88.92±0.1 | 79.25±0.1/82.57±0.1 |
| ALBERT-base+AA-OT | 87.9±0.2 | 54.6±0.5 | 77.0±0.4 | 85.7±0.2 | 91.2±0.1 | 91.3±0.2 | 92.8±0.3 | 91.2 ±0.3 | 81.13±0.1/88.89±0.2 | 79.18±0.1/82.48±0.1 |
| ALBERT-base+AA-CT | **88.6**±0.4 | **55.9**±0.3 | 77.2±0.3 | **85.9**±0.2 | **91.3**±0.1 | **91.5**±0.3 | **93.1**±0.2 | **91.5**±0.2 | **81.32**±0.2/**89.02**±0.1 | **79.33**±0.1/**82.71**±0.1 |

### B.1.3 Model Specifications for Domain Generalizations

We include the results of Decomposable Attention (DA) [57] and Enhanced Sequential Inference Model (ESIM) [58] as baselines from the open-source implementations AllenNLP [82]. Following the setting in Desai and Durrett [51], we also include bert-base-uncased [4] and roberta-base [6] as the pretrained baseline models from HuggingFace Transformers [47]. For BERT, the finetune epoch is 3, batch size is 32, learning rate is $2e^{-5}$, gradient clip is 1.0, and no weight decay. For RoBERTA, the finetune epoch is 3, batch size is 32, learning rate is $1e^{-5}$, gradient clip is 1.0 and weight decay is 0.1. The optimizer is AdamW [83].

### B.1.4 Experimental Settings for Domain Generalizations

Following the settings in Desai and Durrett [51], we test domain generalization on three challenging tasks: (1) *Natural Language Inference.* The Stanford Natural Language Inference (SNLI) corpus is a large-scale entailment dataset [52]. Multi-Genre Natural Language Inference (MNLI) [53] has the similar entailment data across domains. The MNLI can be seen as out-of-domain test dataset. (2) *Paraphrase Detection.* Quora Question Pairs (QQP) contains semantically equivalent sentence pairs from Quora [54]. TwitterPPDB (TPPDB) is considered as out-of-domain data which contains the sentence pairs from the paraphrased tweets [55]. (3) *Commonsense Reasoning.* Situations With Adversarial Generations (SWAG) is a grounded commonsense reasoning task [56]. HellaSWAG (HSWAG) is out-of-domain data which is a more challenging benchmark [56].

### B.1.5 Adversarial Robustness

For the adversarial attack, we follow the setting from Morris et al. [60] and utilize the same models and training procedures as the in-domain natural language understanding. The maximum sentence length is 512.

## B.2 Graph Neural Networks

### B.2.1 Model Specifications

Following the setting in Veličković et al. [64], we use the two-layer GAT model. Models are initialized with Glorot initialization [84] and trained with the cross-entropy loss using the Adam SGD optimizer [85] with an initial learning rate of 0.01 for Pubmed, and 0.005 for all other datasets.

### B.2.2 Detailed Experimental Settings

We follow the architecture and hyperparameters settings in [64]. The number of attention head is 8 in the first layer computing 8 features each followed by an exponential linear unit (ELU) [86] nonlinearity. The second layer is a single-head attention for classification. Dropout [87, 88] is set as $p = 0.6$ and is applied to both layers' input and normalized attention coefficients. In addition, we apply $L2$ regularization with $\lambda = 0.0005$ during training. Pubmed required slight changes to the architecture. The second layer has 8 attention heads and the weight $\lambda$ of $L2$ regularization is 0.001. Early stopping strategy on both the cross-entropy loss and accuracy on the validation nodes are adopted for Cora, Citeseer and Pubmed [65]. The patience is 100 epochs.

## B.3 Visual Question Answering

### B.3.1 Model Specifications

We use the state-of-art VQA models, MCAN [68] which consists of MCA layers. Two types of attention in the MCA layer are self-attention (SA) over question and image features and guided-attention (GA) between question and image features. Mult-head structure is included in each MCA layer with the residual and layer normalization components. By stacking multiple MCA layers, MCAN gradually extract the image and question features through the encoder-decoder structure. Four co-attention layers' MCAN is used in our experiment.

### B.3.2 Experimental Settings

We conduct experiments on the VQA-v2 dataset [67], consisting of human-annotated question-answer pairs for images from the MS-COCO dataset [89]. The whole dataset is split into the three parts. For training, there are 40k images and 444k QA pairs. For validation, there are 40k images and 214k QA pairs. For testing, there are 80k images and 448k QA pairs. The evaluation is conducted on the validation set as the true labels for the test set are not publicly available [90]. For the noisy dataset, we perturb the input by adding Gaussian noise (mean 0, variance 1) to the image features [69]. We use the same model hyperparameters and training settings in Yu et al. [68] as follows: the dimensionality of input image features, input question features, and fused multi-modal features are set to be 2048, 512, and 1024, respectively. The latent dimensionality in the multi-head attention is 512, the number of heads is set to 8, and the latent dimensionality for each head is 64. The size of the answer vocabulary is set to $N = 3129$ using the strategy in Teney et al. [91]. To train the MCAN model, we use the

Adam optimizer [85] with $\beta_1 = 0.9$ and $\beta_2 = 0.98$. The base learning rate is set to $\min(2.5te^{-5}, 1e^{-4})$, where $t$ is the current epoch number starting from 1. After 10 epochs, the learning rate is decayed by $1/5$ every 2 epochs. All the models are trained up to 13 epochs with the same batch size of 64.

### B.3.3   Ablation Study

Table 9: Ablation study of alignment-weight hyperparameter on VQA.

| | ACCURACY ↑ | | PAVPU ↑ | |
| --- | --- | --- | --- | --- |
| | ORIGINAL | NOISY | ORIGINAL | NOISY |
| $\lambda = 1$ | 66.98 | 64.55 | 72.15 | 69.95 |
| $\lambda = 0.1$ | 67.00 | 64.54 | 72.18 | 69.93 |
| $\lambda = 0.01$ | **67.01** | **64.57** | **72.21** | **69.98** |