# OpenReview forum: "Alignment Attention by Matching Key and Query Distributions"
_NeurIPS.cc/2021/Conference — NeurIPS 2021 Poster_

### Official Review · Reviewer_UFmi · 2021-07-17

**Rating:** 6
**Confidence:** 4

**Summary:**

This paper aims to improve the performances of current widely-used attention models by explicitly aligning the query and key distributions. To this end, the authors propose an alignment attention model, which can be easily incorporated into existing self-attention based models, including the pre-training models, e.g., BERT, to boost their performances while remaining efficient in memory and computation cost. To prove the effectiveness of the proposed approach, the authors conduct extensive experimental studies on natural language understanding tasks, graph attention network, and visual question answering task. The domain generalization ability and adversarial robustness of alignment attention are also studied.

> After Rebuttal:

Thanks the authors for the response. I'd like to keep my initial score.

**Limitations And Societal Impact:**

No.

- If the approach brings errors, and what type of errors does it bring, and why? More importantly, are there errors caused by the approach when the model well aligns the distributions of query and key?
- The margins of the gains in all Tables are small and there appears to be no statistical significance testing. Statistical significant test and error ranges are highly appreciated.

**Main Review:**

Strengths:
1. The paper is well written.
2. The paper gives extensive experiments to prove the effectiveness of the proposed approach in terms of various aspects, i.e., accuracy, uncertainty estimation, generalization across domains, and robustness to adversarial attacks.
3. The proposed approach achieves state-of-the-art results on several tasks while remaining efficient in memory and computation cost.

Weaknesses:
1. The contributions are limited.
- The idea of alignment learning has been explored in literature. Thus, the main contribution of this paper is the methodological contribution, i.e., the alignment attention model, which can align the query and key distributions. However, the motivation is unclear.

2. The motivation and the studied problem are unclear.
- Personally, it is unclear what problems of the conventional self-attention model can be solved by the proposed approach? In other words,  what are the problems that will cause if the distributions between query and key are not aligned?
- Specifically, in L44-45, the authors claim that "misalignment could lead to an undesired dependence between the attention weights of a token and its position in the input feature space", however, there is no evidence/experimental analysis that can support this claim. How to prove it? Although the extensive experiments have indeed demonstrated the effectiveness of the proposed approach, I would like to know why performing the alignment of query and key distributions can improve the performance.

3. Some detailed analyses are missing.
- Can the proposed alignment attention be incorporated into the source-to-target attention component?
- In the discriminator-based modules, i.e., L137, why do you adopt the complex highway architecture to construct the discriminator? What are the performances of a single MLP (i.e., FC-Relu-FC) or a fully connected layer (i.e., FC)? Besides, to train the discriminator, what are the output dimension in the discriminator and the adopted loss, e.g., BCE/MSE?
- In Section 2.3, what are the performances of adopting the widely-used distance losses, e.g., KL Loss, to perform the alignment.
- In Equation (2), what are the performances of exchanging query and key distributions?
- If the approach brings errors, and what type of errors does it bring, and why? More importantly, are there errors caused by the approach when the model well aligns the distributions of query and key?
- The margins of the gains in all Tables are small and there appears to be no statistical significance testing. Statistical significant test and error ranges are highly appreciated.

4. Some parts of the paper can be made clearer for easier reading.
- In all Tables, I encourage the authors to list several performances of recently published models to help the readers better understand the proposed approach.
- For the results of all base models in all experiments, it is better to clarify whether the results are reproduced by yourself or copied directly from the original paper.
- L48: 'emprical' -> 'empirical'; L180: 'is' -> 'are'; L269: 'with maximum' -> 'with a maximum'.

I highly encourage the authors to address these concerns and I am glad to increase my score if I see satisfying answers, thanks!


**Time Spent Reviewing:**

7 hours

---

> ### Author Response · Authors · 2021-08-10
> **Response to Reviewer UFmi**
>
>
> Thank you for your insightful and detailed comments and suggestions. Below please find our response.
>
>
> >Q1: Contributions
>
> - We believe the concept of alignment attention is novel. Previous discriminator-based, optimal transport-based, and conditional transport-based methods all typically focus more on generative modeling, where the goal is to align the generated distribution with the observed data distribution. In alignment attention, the goal is to align two different sets of features linearly projected from the same set of observed input features, where both the query and key features need to be adjusted during the learning with alignment attention.
>
>
> >Q2: Query and key alignment motivation
>
> - First, in the traditional attention module, the aggregation between keys and queries is through the dot product, which introduces the implicitly regularized alignments between keys and queries.
> - Second, in comparison to dot-product, our alignment is an explicit regularization. The dot-product focuses on pairwise matching from points (words) to points (words). Our proposed method takes all word representations within the head as discrete empirical distributions and aligns between these two discrete distributions.
> - Third, the linear projections of query and key may not imply they will be well-aligned. For example, after linear projections, a circle becomes one oval with 45 degrees and another oval with 135 degrees. After the inner product, many points on these two ovals may become 0. More specifically, if [0, 1] becomes [1,1] after the key linear projection and [-1,1] after the query linear projection, then the dot product between the key and query would be 0.
> - Fourth, line 41-45 includes our original intuition that motivates us to develop the alignment attention, where we suspect misalignment might lead to undesired dependence between the attention weights of a token and its position in the input feature space (e.g., less frequent but semantically and sentimentally important words may be biased towards smaller attention weights).
>
>   - As we find it is difficult to construct proper experiments to explicitly verify that original intuition, throughout the paper, we have instead focused on experimental evidence to verify the effectiveness of alignment attention motivated by that original intuition.
>
>   - The results of the extensive experiments indeed show better query and key distribution matching after the alignment, as confirmed by the visualization in Figure 1, smaller MMD (MMD is 0.0038 with the alignment loss vs. 0.057 without the alignment loss on MRPC task; please see our response to Reviewer qa5u for more details), and smaller angle (with alignment: mean: 0.6180, std: 0.1749 vs. without alignment: mean: 0.7064, std: 0.1980).
>
>   - We also note that the comparison between Figures 1(c) and 1(d) seems to suggest that alignment attention could help better attend to  semantically and sentimentally important (but overall less frequent) words.
>
>
> >Q3: Can it be incorporated into the source-to-target attention?
>
> - This appears to be a very interesting topic to investigate. Our current belief is that whether this could be beneficial is task dependent. For tasks where the source and target domains are closely coupled, such as domain adaptation, this alignment attention could be very useful, whereas for tasks with distinct source and target domains, such as Visual Question Answering, this alignment attention may not be meaningful. We leave this interesting topic for future study.
>
> >Q4: Details of discriminator
>
> - We include the highway architecture in the discriminator as it is commonly used in sequence-related adversarial models such as [Ref1, Ref2]. Following your suggestion, we have tried a single MLP (FC-Relu-FC) and achieved consistent improvements on GLUE data as:
>
>   - MRPC: 87.4, COLA: 55.7, RTE: 77.2, MNLI: 85.5, QNLI: 91.3, SST: 92.5, STS: 91.1.
>
>  -
> The output dimension of the discriminator is [batch size, number of head, number of query or key, 1] and we adopt the BCE as the loss.
>
> >Q5: KL loss
>
> - The proposed alignment loss is defined over two discrete distributions whose supports are not overlapped. Therefore, some widely used statistical distances, such as the KL divergence, are not applicable in our alignment attention setting.
>
>
> >Q6:Exchanging query and key
>
> - In Equation (2), there is no difference between exchanging query and key in theory.
>
> >Q7: Results of base models
>
> - We will clarify whether the results are reproduced or quoted directly from the original paper and add these details in the revision.
>
> >Q8: What type of errors does the proposed approach bring, and why?
>
> - The same as other regularization techniques[Ref3, Ref4], alignment attention could lead to performance drop when the regularization is too strong.
> For example, in the ablation study to exam the role of the alignment-weight hyperparameter $\lambda$ in Equation 1, we turn it from 0.01 to 1. We find that the experimental results are not sensitive to the choice of  $\lambda$ within this range. Any number from 0.01 to 1 would give similar results. The detailed results are in Table 9 in Appendix. However, if we enforce the alignment weight to be much larger, such as $\lambda=100$, it could  severely over regularize the query and key distribution, forcing the network to overly focusing on matching the two distributions and hence clearly hurt the empirical performance.
>
> >Q9: Error bars
>
> - The sample mean and error bar of Table 1 were deferred to Table 8 in the Appendix. We included error bars of our method in Table 4 and Table 5 (we omitted some error bars, trying to prevent the table from becoming too crowded. We will add these omitted error bars to the revision). The provided error bars show that the improvements are statistically significant.
>
>
> **References:**
>
> [Ref1] Yu, Lantao, Weinan Zhang, Jun Wang, and Yong Yu. "Seqgan: Sequence generative adversarial nets with policy gradient." In Proceedings of the AAAI conference on artificial intelligence, vol. 31, no. 1. 2017.
>
> [Ref2] Guo, Jiaxian, Sidi Lu, Han Cai, Weinan Zhang, Yong Yu, and Jun Wang. "Long text generation via adversarial training with leaked information." In Proceedings of the AAAI Conference on Artificial Intelligence, vol. 32, no. 1. 2018.
>
> [Ref3] Gal, Yarin, and Zoubin Ghahramani. "Dropout as a bayesian approximation: Representing model uncertainty in deep learning." In international conference on machine learning, pp. 1050-1059. PMLR, 2016.
>
> [Ref4] Gal, Yarin, Jiri Hron, and Alex Kendall. "Concrete dropout." arXiv preprint arXiv:1705.07832 (2017).

---

> > ### Comment · Reviewer_UFmi · 2021-08-30
> > **Re: Response to Reviewer UFmi**
> >
> > Thanks the authors for the response. I'd like to keep my initial score.

---

> > > ### Author Response · Authors · 2021-08-30
> > > **Thank you**
> > >
> > > We greatly appreciate your effort in reading our response! We will carefully revise our paper according to your feedback.

---

### Official Review · Reviewer_E8gR · 2021-07-21

**Rating:** 5
**Confidence:** 5

**Summary:**

The paper reviews the attention mechanism commonly used in deep neural networks. It proposes the alignment attention mechanism that match the query and key distributions in self-attention within each head. The paper regards the alignment attention as an unsupervised regularization in current attention. Three alignment methods have been introduced to explore the effectiveness of distribution matching. Extensive experiments are done to demonstrate the effectiveness and general applicability of alignment attention in domains of language understanding tasks, and even graph node classification and visual question answering. The alignment attention shows great potential of being a plug-and-play component to current attention models.

**Main Review:**

Main Review:

The paper rethinks the dominant self-attention mechanism in both deep neural networks and points out the alignments of query and key distributions in the current attention mechanism are largely unexplored. The motivation of align the projection of both query and key tokens is strong and meaningful as decreasing the risk of misaligned interactions between queries and keys. To conclude, the paper attempts to take distribution matching into consideration of attention mechanism.
Based on that motivation, the authors proposed three alignment methods attempting to align the attention and finally adopted adversarial training and conditional transport methods to align the distributions. The analyses of methods are theoretical and insightful.
The authors conducted extensive experiments in natural language understanding, GNN and VQA domains. The evaluations are not only in-domain dataset, but also out-domain ones to illustrate the generalization across domains. Adversarial attacks are taken into consideration to show the robustness of alignment attention. The experiments are very meticulous and rational to show the effectiveness of alignment attention as a plugin-and-play component.

Limits:

As is known, the multi-head attention mechanism is widely applied in vision tasks like classification, detection and segmentation. The authors have conducted detailed ablations to show effectiveness in NLP, GNN and VQA covering language, graph and multi-modal domains. I suggest supplements of some vision tasks to finalize the ablations would make the paper more completed

Reason for weak rejection:

My concern about the paper is that do we really need alignment between key and query? As key and query are linear projections of input, the distribution of key and query should be well aligned. Thus I do not think the alignment story is reasonable and suggest weak rejection.


**Time Spent Reviewing:**

10 hours

---

> ### Author Response · Authors · 2021-08-10
> **Response to Reviewer E8gR**
>
> Thank you for your thoughtful comments and suggestions. Please find our response below.
>
> >Q1: Alignment attention for vision tasks
>
> Given the surge of recent interest in vision Transformers and related models in vision tasks, we agree that adapting our approach to vision tasks will be exciting to explore! We have already observed promising results in language understanding tasks, domain generalization, adversarial attacks, graph node classification, and visual question answering.
> Following your suggestion, we have further conducted a preliminary evaluation of incorporating our alignment method into DeiT: Data-efficient Image Transformers (Hugo Touvron, Matthieu Cord, Matthijs Douze, Francisco Massa, Alexandre Sablayrolles, and Hervé Jégou. "Training data-efficient image transformers & distillation through attention." In ICML, 2021.). On DeiT-tiny, the baseline achieves 71.9 on acc@1 on Imagenet2012, while the discriminator-based alignment attention and bi-directional conditional transport-based one achieve 72.7 and 72.5, respectively, on acc@1.
> We will include more discussions about potential applications of alignment attention in vision tasks in the revision.
>
> >Q2: Alignment between key and query
>
>
> We hope to clarify that using linear projections does not necessarily imply the keys and queries will be well-aligned. Suppose a circle becomes an oval with 45 degrees after a linear projection and an oval with 135 degrees after another linear projection. After the inner product, many points on these two ovals may become 0. For example, if [0, 1] becomes [1,1] or [$-$1,1] after two individual linear projections, the dot product between these two linear projections becomes 0. Our alignment method can discourage these by introducing explicit alignment regularization between queries and keys. It takes all word representations within the head as the random samples of two different distributions and aligns between these two distributions. We will include more discussion in the revision.

---

> > ### Comment · Reviewer_E8gR · 2021-09-01
> > **Final Review**
> >
> > Thanks for your kind response and additional experiments.
> >
> > However, I keep my score unchanged as I do not think the idea of matching between key and query distribution is quite reasonable. The gain may not come from distribution matching between key and query. Adversarial training has been shown to improve the performance of BERT/Transformer. For example, FreeLB[1] shows that adding adversarial perturbation can improve the performance of the BERT or transformer-based model. The author is highly encouraged to design better experiments to analyze the reasons for improved performance.
> >
> > [1] Freelb: Enhanced adversarial training for natural language understanding

---

> > > ### Author Response · Authors · 2021-09-02
> > > **The role of adversarial learning in FreeLB is different from that in our paper; they are complementary rather than conflicting**
> > >
> > > We appreciate your providing additional feedback. We hope our response below could help convince you to reconsider the significance of our contributions.
> > >
> > > 1. We'd like to emphasize that adversarial is only one of the methods for aligning distributions. We have presented the optimal transport (OT) based alignment, which does not introduce adversarial learning at all. As shown in Table 1, the OT-based alignment methods also provide consistent improvements.
> > >
> > > 2. Moreover, the proposed adversarial learning, which is used to align distributions, perturbs neither the word embeddings nor the hidden units of other layers.
> > >
> > > 3. Thank you for recommending FreeLB, which adds adversarial perturbations to the word embeddings. By contrast, our alignment attention explicitly encourages self-attention to match the distributions of the key and query within each hidden layer, where either an adversarial learning or optimal transport-based method could help encourage distribution alignment. Note there is no adversarial perturbation in the proposed alignment attention. Therefore, our method has no conflict with FreeLB at all, and hence it is possible to combine both to let them strengthen each other. We will elaborate on this point in the revision.

---

> ### Author Response · Authors · 2021-08-30
> **Thanks for the reviews**
>
> Dear Reviewer E8gR, please let us know if you have any questions or concerns that you'd like us to explain or clarify. Thanks!

---

### Official Review · Reviewer_qa5u · 2021-07-25

**Rating:** 7
**Confidence:** 4

**Summary:**

This work aims at solving the alignment problem between the query and key distribution within the self-attention module. It is based on the hypothesis that the key and query following identical distribution can help to improve the feature learning for the attention module.
Correspondingly, an alignment regularization is proposed to explicitly encourage self-attention to match the key and query within each head. Three kinds of alignment methods, including Adversarial training-based, Optimal transport-based, and, Bidirectional conditional transport-based, all of which can outperforms the baselines. Comprehensive experiments are performed on a variety of problem settings, which proves that the introduced alignment module can bring consistent improvements. Besides, the proposed method can work as a plug-and-play module to be added to many existing attention modules.

**Main Review:**

I recommend accepting the paper. In general, this paper is well-written and easy to follow. It aims at studying the interactions between the keys and queries for attention modules. A variety of experiments on different problem settings with different proposed alignment methods are performed, and, the proposed method consistently outperforms the baseline, which proves that explicitly regularizing the key and query distributions can help to improve the feature learning for the attention-based methods.  I summarize the strengths and weaknesses as follows:

Strengths:
- This paper proposes a plug-and-play alignment module that can be applied to various attention-based models.
- This paper proves that explicitly regularizing the distributions of the key and query feature can help the feature learning of the self-attention module.
- Three alignment methods are introduced to regularizing the distributions of the keys and queries.
- The experimental results are convincing. Experiments are performed in different language understanding settings. It shows that the proposed alignment can bring consistent improvements no matter which alignment method is adopted.

Weaknesses:
- It lacks the experimental results showing that after being trained with the alignment loss, the distribution of the keys and queries are well-aligned.
- In the baseline attention module, the keys and queries will be dot-producted, which means they should be of the same feature domain. Thus, it has implicitly introduced the regularization of their alignments. In comparison, the novel part of the proposed method is explicitly modeling the regularization. However, it lacks a detailed explanation on why explicitly align the keys and queries can bring improvements over the baselines.
- Typos: Line-104 the tensor dimension should be [B, H, w, d]

**Time Spent Reviewing:**

3h

---

> ### Author Response · Authors · 2021-08-10
> **Response to Reviewer qa5u**
>
> Thank you for your valuable feedback. Below please find our response to the list of questions pointed out in the review.
>
> >Q1: Results after being trained with the alignment loss
>
> Figure 1 provides the visualization of the distributions of the keys and queries with and without the alignment loss. To further confirm that the distribution of the keys and queries are well-aligned after training with alignment loss, we use Maximum Mean Discrepancy (MMD) with standard Gaussian kernels to measure key and query distribution discrepancy and compare MMD with or without alignment loss. Aggregating the MMDs across all heads and layers,  on Microsoft Research Paraphrase Corpus (MRPC) task, the total MMD with the alignment loss is 0.0038, while that without the alignment loss is 0.057. This confirms our conjecture from visualization. We will include these numerical results and strengthen the discussion of this in the revision.
>
>
> >Q2: Implicit versus Explicit regularization
>
> In comparison with dot-product, our alignment attention is not only an explicit regularization, but also a distribution-wise regularization instead of a pair-wise regularization. While the dot-product focuses on pairwise matching from token to token, our proposed method takes all key and query embeddings within each head as the random samples of two different distributions and aligns between these two distributions, which is shown to produce a better regularization of the alignments.
>
> >Q3: Tensor dimension
>
>  We will clarify both Q and K have dimensions [B, H, w, d] and the point-to-point difference from query to key is of dimension [B, H, w, w].

---

### Decision · Program_Chairs · 2021-09-27

**Decision:**

Accept (Poster)

**Comment:**

I recommend acceptance of the paper for the following reasons. As pointed out by reviewer E8gR, the premise of aligning key and value distributions might not be as important as it sounds at first for the purposes of improved learning. In addition, the analysis and ablations studies could be more thorough and convincing. Yet, experimentally the module works well. So despite potential limitations on the "intuition" and "analysis" of the module, there is value in knowing that such a module has empirical validation, especially for its potential impact on future analyses. There is also agreement from reviewers about this point and general support from two reviewers for the paper.

For these reasons, I recommend to accept the paper.